# On the Geochemistry of Major and Trace Elements Distribution in Sediments and Soils of Zarafshon River Valley, Western Tajikistan

Djamshed A. Abdushukurov [1], Daler Abdusamadzoda [1,2], Octavian G. Duliu [2,3,4,*], Inga Zinicovscaia [2,5] and Pavel S. Nekhoroshkov [2]

1 Institute of Water Problems, Hydropower and Ecology of Academy of Science, 14a Ainy Str., Dushanbe 734042, Tajikistan; abdush_dj@mail.ru (D.A.A.); martinez-91@mail.ru (D.A.)
2 Joint Institute for Nuclear Research, 6, Joliot Curie Str., 141980 Dubna, Russia; zinikovskaia@mail.ru (I.Z.); p.nekhoroshkov@gmail.com (P.S.N.)
3 Department of Structure of Matter, Faculty of Physics, University of Bucharest, Earth and Atmospheric Physics and Astrophysics, 405, Atomistilor Str., 077125 Magurele, Romania
4 Geological Institute of Romania, 1, Caransebes Str., 012271 Bucharest, Romania
5 Department of Nuclear Physics, Horia Hulubei National Institute for R&D in Physics and Nuclear Engineering, 30 Reactorului Str., P.O. Box MG-6, 077125 Magurele, Romania
* Correspondence: o.duliu@upcmail.ro

**Abstract:** To assess the geochemical features of sedimentary material of Zarafshon river, (Western Tajikistan) catchment basin, the mass fractions of 38 major and trace elements were determined by Instrumental Neutron Activation Analysis (INAA) in 2 × 116 paired samples of sediments and soils collected along the Zarafshon River and its main tributaries from the sources to Tajik—Uzbek border. At each collecting location, the distance between sediments and soils' sampling was no greater than 10 m allowing the studying of the interrelation between sediments and soils. This evidenced a significant similarity between paired soils and sediments' samples, including the potentially contaminating elements As, Sb and Hg, whose mass fractions in some places were significantly higher than for the Upper Continental Crust (UCC) and North American Shale Composite (NASC), suggesting a common provenience. At the same time, the distribution of major, as well as of incompatible trace elements, Sc, Zr, REE, Th, and U, in spite of geological diversity of the Zarafshon river catchment basin, suggest a possible felsic origin of investigated material.

**Keywords:** Zarafshon; Tajikistan; sediments; soils; INAA; felsic material; REE

## 1. Introduction

The Zarafshon River originates at the Zarafshon glacier, at an altitude of 2775 m, at the junction of the Turkestan and Zarafshon ranges, both of them belonging to the western sector of the Pamir-Alay system [1]. The initial part of the river, about 300 km long, lies in a narrow, deep valley. On the southern left bank, flowing between the Turkestan and Zarafshon ranges, it receives the Yagnob, Arthuch and Mogiyon rivers, as well as many small tributaries (Figure 1) [1–3]. After the city of Panjakent, the river follows a flat region, crosses the Tajikistan-Uzbekistan border, passes Samarkand and Bukhara, to finally disappear in the desert in the vicinity of the city of Karakul, without reaching the Amu Darya River [3,4]. Its catchment basin, with an area of 17,700 km$^2$, covers diverse geological formations spread from the alpine zone, 3200–3500 m above sea level, to less than 150 m elevation westward towards the city of Karakul. In the Zarafshon mountain gorges, below 1500 m there are small areas of semi-savannahs as well as mountain forests. Here, regosols are prevalent [5], while in the western sector of Zarafshon Valley luvisols formed on loess deposits are common. Moreover, in mountain gorges, terraces of alluvial

deposits and outflow cones of lateral tributaries serve as the basis for soil components (Figure 1a, Table A1).

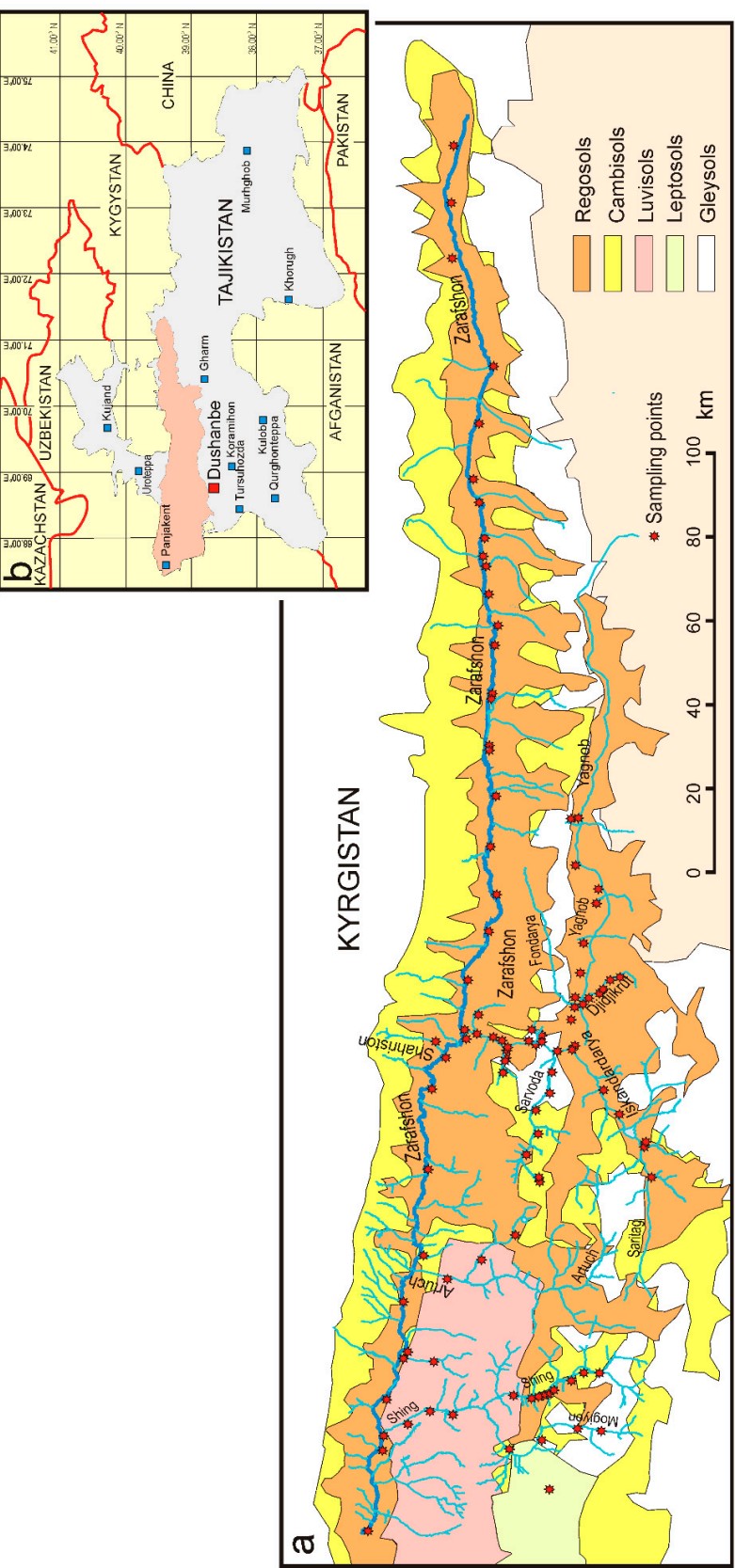

**Figure 1.** The map of Zarafshon Valley (**a**); showing its position within Tajikistan territory (**b**).

The "Zarafshon" name, which in Farsi means "spreader of gold", reflects the existence of ore fields along the river valley which, together with other outcrops of nonferrous ores makes it a mineral rich province requiring a detailed investigation.

Although the total population of Zarafshon valley does not exceed 400,000 inhabitants, mainly populating small villages, the existence of non-ferrous ore outcrops could represent the main source of down-stream contamination, previously analyzed in [6].

For this reason, the distribution of major and rock-forming elements as well as of incompatible lithophile elements such as Sc, Ti, V, Cr, Zr, lanthanides (LN) Th or more soluble U together with siderophile Co and Ni were chosen for this study. All of them play a significant role in geochemistry and pedology as tracers of the origin of natural material which, disintegrated during weathering and erosion, represents the main constituents of sediments and soils [7–10]. Besides them, there are other trace elements such as V, Cr, Co, Ni, Zn, As, Sb, or Hg, intensively used in diverse industrial processes, and, the presence of which, at levels significantly exceeding the natural ones, are related to an anthropogenic contamination which could be industrial [11,12] or due to urban as well as rural agglomerations [13]. According to [14], soil is a result of the combined action of weather, geographic relief, organisms and human activity on the parent mineral substrate, all of them continuously interacting. The actual soil is no older than Pleistocene, i.e., 2.59 Ma almost coincident with the onset of Northern hemisphere glaciation.

In turn, sediments represent a fragmented mineral substrate under the action of weathering and erosion and which is transported at long distances by the action of water, wind or ice and finally deposited. With respect to soil, sediments consist of unconsolidated clastic material with a reduced amount of organic matter. When consolidated by natural cementation processes, sediments turn into sedimentary rocks such as siltstone, sandstone, shale, etc.

Both sediments and soils preserve the fingerprint of the parental mineral substrate, which, under the action of external factors, evolve in a different manner. Collected from the same places and investigated in tandem, these sedimentary materials could bring a bonus of information concerning the local geochemistry as well as their interrelationship.

In this regard, in the period of spring-autumn, between 2015 and 2018, the Institute of Water Problems, Hydropower and Ecology of the Academy of Science of Tajikistan organized a field expedition to collect the most representative samples of sediments and soils from the Tajik sector of Zarafshon catching basin. For a better characterization of the soil–sediments interrelation, we collected from the same places distributed along the Zarafshon River and its main tributaries, including both sediments and soils samples (Figure 1a). In the case of presumed contaminated locations, mainly nonferrous mines and processing plants, samples were collected upstream and downstream at distances of no more than 10 to 20 m from them.

This permitted the collecting of 116 sediments, as well as an equal number of soils samples to be analyzed at the Neutron Activation Analysis Section of the Frank Laboratory of Neutron Physics (FLNP) of the Joint Institute for Nuclear Research (JINR), Dubna, Russian Federation. For a better precision and accuracy, we have used the Instrumental Neutron Activation Analysis (INAA), in both its thermal and epithermal variants [15,16]. It is worth mentioning that INAA was chosen due its capacity to determine the mass fractions of up to 40 elements, 38 in the case of this study, without any preliminary processing prone to induce systematic errors (Table 1) [16].

**Table 1.** The mass fraction of investigated elements in sediments and soils as well as the main literature references: UCC [17]; North American Shale Composite NASC [18]; Post Archaean Australian Shale (PAAS) [8]; Suspended Matter of the World's Rivers (SMWR) [19]; as well as Average World Suspended Sediments (AWSS) [20]. Oxides' mass fraction expressed in wt.%, in mg/kg for other elements, excepting Hg of which the mass fraction is expressed in µg/kg. CSU represents the Combined Standard Uncertainty [21].

| Element | Sediment | CSU | Soils | CSU | UCC | NASC | PAAS | SMWR | AWSS |
|---|---|---|---|---|---|---|---|---|---|
| $SiO_2$ | 62.75 | 12.65 | 59.39 | 9.7 | 66.62 | 64.48 | 62.8 | 54.76 | 54.34 |
| $TiO_2$ | 0.62 | 0.22 | 0.83 | 1.06 | 0.64 | 0.72 | 1 | 0.65 | 0.73 |
| $Al_2O_3$ | 9.9 | 3.52 | 12.92 | 4.07 | 15.4 | 16.9 | 2.2 | 16.31 | 16.48 |
| FeO | 3.79 | 2.5 | 5.78 | 2.01 | 5.04 | 5.7 | 18.9 | 6.47 | 7.47 |
| $MnO_2$ | 0.07 | 0.07 | 0.09 | 0.03 | 0.1 | 0.06 | 0.11 | 0.15 | 0.22 |
| MgO | 6.32 | 3.42 | 5.8 | 2.72 | 2.48 | 2.85 | — | 2.39 | 2.09 |
| CaO | 10.59 | 10.71 | 9.46 | 9.06 | 3.59 | 3.56 | 1.3 | 3.64 | 3.62 |
| $K_2O$ | 2.4 | 0.99 | 2.83 | 0.79 | 2.8 | 3.99 | 3.7 | 2.59 | 2.04 |
| $Na_2O$ | 1.4 | 0.79 | 0.9 | 0.41 | 3.27 | 1.15 | 1.2 | 1.11 | 0.96 |
| Sc | 9.6 | 4 | 11 | 5 | 14 | 15 | 16 | 14 | 18 |
| V | 101 | 56 | 254 | 1500 | 97 | — | — | 120 | 129 |
| Cr | 71 | 40 | 90 | 57 | 92 | 124 | 110 | 85 | 130 |
| Co | 11.5 | 6.4 | 14.5 | 7.1 | 17.3 | — | — | 19 | 22.5 |
| Ni | 46 | 32 | 65 | 31 | 47 | 58 | 55 | 50 | 74 |
| Zn | 102 | 57 | 119 | 54 | 67 | — | —- | 130 | 208 |
| As | 63 | 254 | 84 | 600 | 4.8 | 28.4 | — | 14 | 36.3 |
| Br | 2.6 | 4.3 | 1.9 | 1.6 | 1.6 | 0.9 | — | 9 | 21.5 |
| Rb | 92 | 41 | 103 | 39 | 84 | 126 | 160 | 77.0 | 79 |
| Sr | 143 | 63 | 177 | 100 | 320 | 142 | 200 | 150.0 | 187 |
| Zr | 220 | 133 | 210 | 101 | 193 | 200 | 210 | 150.0 | 160 |
| Sb | 59 | 560 | 38 | 270 | 0.4 | — | — | 1.4 | 2.2 |
| Cs | 4.5 | 570 | 7.5 | 3.6 | 4.9 | 5.2 | 15 | 5.2 | 6.3 |
| Ba | 724 | 450 | 784 | 500 | 624 | 636 | 650 | 500 | 522 |
| La | 28 | 14 | 32 | 12 | 31 | 31 | 38.2 | 32 | 37 |
| Ce | 53 | 26 | 61 | 25 | 63 | 67 | 79.6 | 68 | 73.6 |
| Nd | 27 | 15 | 32 | 17 | 27 | 27 | 33.9 | 29 | 32.2 |
| Sm | 5 | 2.5 | 6.3 | 3.4 | 4.7 | 5.6 | 5.6 | 5.8 | 6.1 |
| Eu | 0.9 | 0.6 | 1.1 | 0.8 | 1 | 1.2 | 1.1 | 1.4 | 1.3 |
| Gd | 3.9 | 3.3 | 2.9 | 2.4 | 4 | — | 4.7 | 5.6 | 5.3 |
| Tb | 0.7 | 0.3 | 1 | 0.4 | 0.7 | 0.9 | 0.8 | 0.8 | 0.8 |
| Tm | 0.3 | 0.3 | 0.5 | 0.5 | 0.3 | — | 0.4 | 0.4 | 0.4 |
| Yb | 1.9 | 1.2 | 2.2 | 1.5 | 2 | 3.1 | 2.8 | 2.5 | 2.1 |
| Hf | 5.5 | 3 | 5.8 | 2.6 | 5.3 | 6.3 | — | 4.4 | 4 |
| Ta | 0.8 | 0.4 | 1 | 0.5 | 0.9 | 1.12 | 14.6 | 0.9 | 1.3 |
| W | 2.5 | 5.7 | 2.8 | 2.7 | 1.9 | 2.1 | — | 1.4 | 2 |
| Hg | 2.5 | 14.7 | 1.1 | 4.5 | 0.1 | — | — | 0.1 | — |
| Th | 8.9 | 4.3 | 10.9 | 3.5 | 10.5 | 12.3 | 3.2 | 10 | 12.1 |
| U | 3 | 1.5 | 3.9 | 1.7 | 2.7 | 2.7 | — | 2.4 | 3.3 |

## 2. Hypothesis and Research Objectives

As presented before, the Zarafshon catchment basin covers a diversity of geological formations of which the origin dates back to Oligocene. This was when, as mentioned in [22], a Jurassic basin filled with continental sediments, under the pressure exerted by the Hindustan continental plate, and generated the Pamir-Alay system. Therefore, it is to be expected that the materials of which the Zarafshon valley is made up to contain a significant proportion of crustal matter. In time, due to weathering, this tends to be transported by the river and its tributaries as suspended load and deposited on river beds and river banks, as the mineral constituents of soils.

Accordingly, the main goals of this study consist of:

(i) to verify to what extent the geochemistry of depositional material, i.e., sediments and soils, can be related to crustal material, better approximated by UCC [17]

and NASC [18], PAAS [8] in the case of soils, and SMWR [19], as well as AWSS [20] for sediments;

(ii) to evidence any interrelationship between the geochemistry of sediments and adjacent soils, as it is reflected by the distribution of major rock forming, as well as incompatible elements.

It should be noted that we have paid special attention to those elements whose presence is not related to any industrial activities such as mining, as the presence and distribution of the Presumably Contaminating Elements (PCE) on the Zarafshon catchment basin were previously analyzed in [6].

The results of our study performed under these circumstances will be further presented and discussed.

## 3. Geological Setting

Due to its location between two important mountain ranges—Turkestan and Zarafshon—the Zarafshon catchment's basin covers a variety of lithologies of which clastic material represents the main component of sediments and bordering soils.

According to [22], the actual Turkestan and Zarafshon ranges evolved during Oligocene from a Jurassic basin filled with continental, shallow-water marine and lagoonal sediments of which crustal origin seems to influence its present geochemistry. This was the main reason for which we have chosen UCC [17] and NASC [18] as the most appropriate descriptor of investigated material.

At present, Djidjikrut, Fondarya, both tributaries of the Yagnob, as well as Mogiyon and its tributaries have the most important contribution to the mineral composition of Zarafshon sedimentary material [23,24].

The Djidjikrut River flowing in the Ayni district passes through clays, marls, limestones, dolomites and salts, significantly influencing the mineral composition of water and neighboring sediments and soils.

At the confluence with the Zarafshon (Yagnob river), is located the Djidjikrut ore field related to Zarafshon-Gissar mercury-antimony belt, and dating to Middle and Upper Paleozoic.

Antimonite, cinnabar and metacinnabarite are the main minerals of the Djidjikrut ore field, while pyrite and marcasite with an admixture of Co, Ni, Cu, As, Sb, Au, Te and Bi, realgar and orpiment, as well as hematite and sphalerite are common minerals too. For this reason, at the confluence of the Djidjikrut and Yagnob rivers, on the northern slope of the Gissar range, the Anzob mining and processing plant is operating [5] with the final release of Hg-Sb concentrate, an important source of downstream contamination (Figure A1).

The Yagnob River flows westward through the narrow mountain gorge along the southern edge of the Zarafshon mountains, being the first tributary of the Fondarya River. The total length of the Yagnob river is 114 km with a basin area of 1660 km$^2$, while the mineral composition of sediments and soils of its upper reaches is still poorly investigated. Downstream, near the confluence with Fondarya, the Fon-Yagnob coal deposit occupies almost the entire left-bank site between the Djidjikrut and Pasrud tributaries [25]. Here, due to exothermic oxidation of the sub-surface coal layers, rich in pyrite and marcasite, a self-ignition process contaminated neighboring locations. Here, the paralava, a product of complete melting and (partial) recrystallization of local rocks generated by the high temperature attained during underground coal combustion, consists of almost melted sediments enclosing coal seams [26–28]. It is supposed that the underground coal fire may contribute to diagenesis of some REE minerals such as godovikovite, efremovite, maskanite, and so on [27–30].

The Fondarya River with a significant catchment basin of 3230 km$^2$ is formed by the confluence of the Yagnob and Iskandardarya, which then crosses the Zarafshon ridge and flows into the Zarafshon river. In addition, the Fondarya is fed by the waters of the Sarvoda river which originates from the Alovdin lakes, as well as the lower and upper Kumarg tributaries, Chore and Pete (Figure 1a).

It should be noted that in the upper flows of the Iskandardarya river operates the mining and processing plant "Tolko-Gold", which extracts and processes Au, Ag, Hg, and Sb ore as well as fluorite. The mining operation is also carried out at the gold deposit in the Kumargi Bolo gorge. The Chore River gorge is classified as gold-quartz deposits at the Turkestan-Chore zones of Central Tajikistan [31]. This deposit was discovered in 1960 and intensively exploited during 1974–1984 mainly for pyrite and arsenopyrite which represented more than 95% of all ore minerals together with small amounts of antimonites, chalcopyrite, sphalerite, pyrrhotite, and others [32].

The Pete tributary flows between the Pete-Takfon ore field hosting Takfon, Pete, Simich, and other deposits. In the geological structure of this ore field, Paleozoic formations are widely developed, which are represented by Silurian (shales, limestones, dolomites with interlayers of quartzites), Devonian (siliceous and micaceous shales with interlayers of cherts and limestones), Carboniferous (limestones with horizons of clay shales, sandstones) and Cong deposits. To the south of the ore field area, the complex of Paleozoic formations is overlapped by sub-platform sediments of the Mesozoic. The Igneous formations are represented by the Pete intrusions of granodiorites, dikes of granodiorite-porphyry, and lamprophyres. There are also tin and silver, lead, and zinc-containing ores [33].

The Mogiyon river (Figure 1a), with a length of 68 km and a basin area of 1100 km$^2$, is one of the largest Zarafshon tributaries. From the source to the village of Mogiyon in a narrow valley, the river flows in a northern direction, cutting through the Zarafshon ridge with a deep transverse gorge. Further on, having received its largest tributary–the Shing River–it continues to flow in a narrow rapids channel with a width of up to 10 m. Nine kilometers above the city of Panjakent, leaving the gorge and passing through the village of Sujina, it then falls into the Zarafshon River (Figure 1). Between the Shing and Mogiyon interfluve is located the Taror ore field, which belongs to the Western Zarafshon-Gissar metallogenic zone [34].

However, along with gold-sulfide mineralization, silver-tin-base metal mineralization, localized in its upper and middle horizons, is also developed. The silver deposits in the Taror ore field belong to the Mirkhant deposit and its mineralization is predominantly represented by a silver-tin-polymetallic association spatially combined with sulfide, showing the closest correlation between Ag and Sb [27]. It is worth noting that in the Tarror gold ore deposit operates the Joint Venture "Zarafshon" whose main product consists of Ag-Au alloy [35].

In addition, along Zarafshon river and some of its main tributaries there is an active mining and metallurgical industry, based on the extraction and processing of non-ferrous metals. Among them, the Anzob mining and processing plant (AMPP), Taror gold ore plant (TGOP), Upper Kumarg and Kanchoch AGOK and TGOP specialized in the production of antimony-mercury concentrates and, respectively, gold. All of them could be regarded as a potential source of anthropogenic contamination which needs continuous monitoring extended to both the sediments and neighboring soils [35].

In this regard, the investigated Zarafshon valley together with some of its tributaries are the object of controversy between scientists and environmental experts representing governmental and non-governmental organizations. For this reason, it is not entirely correct to assess the environmental state of a particular region without paying a special attention to its geochemical features.

## 4. Materials and Methods

### 4.1. Sampling

An equal number of 116 sediments and 116 soils samples were collected in the investigated area (Figure 1a), using an AMS 12 Multi Stage sediments Sampler (AFMS Inc., American Falls, ID, USA) equipped with tubular glass sampler. Unconsolidated columns of sediment, 20 to 30 cm long, were handpicked from the riverbeds during a high-water period. As at altitudes greater than 2000 m, the soil cover is less developed, so that soil

columns were no longer than 10 cm. For our study we retained the top soil (horizon A), as we were also interested to evidence recent contamination processes [6].

We collected soil samples always nearby the river banks. After collecting, any traces of vegetable matter such as fragments of grass and grass roots, leaves, wood pieces, as well as fragments of rock were carefully removed.

From each sampling point, three sub-samples of collected material, e.g., sediment or soil, covering an area of about 10 m$^2$ were gathered and carefully mixed to form a composite sample. Any possible cross-contamination was avoided by washing the glass sampler before any sampling with a solution of diluted hydrochloric acid with deionized water and soaking them with disposable cotton serves.

To evidence any correlation between sediments' and soils' geochemistry, sediment and soil samples were collected in the same places, with two exceptions, and the distance between the sediments and the soils' sampling location never exceeded 5–10 m (Table 1) and [36]. The same distances separated mines, adits or tunnels of the downstream collecting places.

### 4.2. Samples Preparation

Samples consisting of silty grayish sand without organic detritus were placed into the plastic container with cooled refrigerant and kept at 4 °C. In the laboratory, all samples were dried at 70 °C at constant weight and ground. Finally, the dried material was sieved using a 0.425 mm (42 mesh) sieve, homogenized and sent to the Frank Laboratory of Neutron Physics (FLNP) of the Joint Institute for Nuclear Research (JINR) for further INAA.

Here, 10 g of each sample were homogenized again for 15 min in a planetary ball mill (PULVERISETTE 6, Fritsch Laboratory Instruments GmbH, Idar-Oberstein, Germany) at 400 rpm. To increase the precision and accuracy of measurements, six aliquots of about 0.3 g were selected from each sample and independently analyzed.

### 4.3. INAA Measurements and Quality Control

Before irradiation at the IBR-2 reactor of the JINR FLNP in Dubna, each aliquot was packed in polyethylene bags for the determination of short-living isotopes (three aliquots) and in aluminum foil (three aliquots) for long-living ones. It is worth mentioning that the mass fractions of short-living isotopes were determined by thermal neutron activation while in the case of long-living isotopes we have used epithermal neutrons. More details concerning irradiation time and neutron flux densities in each irradiation channel can be found in [37].

After irradiation, the gamma spectrum of each sample was recorded using a HPGe detector with a resolution of 1.9 keV for the $^{60}$Co 1332 keV line, analyzed by means of a Genie 2000 Canberra software and processed by a proprietary software [37]. This permitted determination of the mass fraction of each of the 38 considered elements together with the associated CSU [21]. In all cases, the CSU was estimated by taking into account the statistical error, influence of measurement geometry, detector efficiency, the content, as well as the error of each element for all Certified Standard Material (CSM) utilized for calibration. To avoid any environmental contamination, all resulting radioactive materials were deposited in a permanent low activity repository.

The quality control was assured in the case of sediments and soils by a simultaneous analysis of Standard Reference Materials (SRM) provided by the National Institute for Standard and Technologies (NIST), i.e., 1633c Coal fly ash, 667 Estuarine sediment, 2710 Montana Soil and 1547 Peach leaves for short half-life time isotopes. In the case of long life-time isotopes, the best results were obtained using–2709 Trace elements in soil, 1632c–Trace elements in coal, 690CC–Calcareous soil, 2709a–San Joaquin soil, 1632c–Trace elements in coal and SRM-AGV2–Andesite.

Further, by using all of the above-mentioned SRM a Group of Standard Sample (GSS) [37,38] was realized which permitted the choice of the most appropriate SRM lines to minimize errors in determining the mass fractions of all investigated elements. It is worth

mentioning that GSS is a proprietary software which, besides helping to determine the mass fractions of chosen elements, it allowed the calculation of the extent to which the mass fractions of SRM corresponded to certified ones [37,38]. Following this procedure, the accuracy quantified by means of standard deviation corresponding to each group of three aliquots was lower than 10%. In the case of SRM, the CSU varied between 3 and 15%, higher in the case of REE, but never greater than 20%. It is worth mentioning that for all 38 investigated elements, the detection limit was of 1 mg/kg and lower [37,38].

To illustrate the sensitivity and accuracy of INAA measurements following the CSS software, in Table A2 we have reproduced, as an example, the experimentally determined mass fractions of SRM 2709. Similar determinations were completed in the case of all utilized SRM.

## 5. Results and Discussion

A complete set of data concerning the mass fractions of all 38 elements in the investigated samples are provided in the Mendeley repository [36], while Table 1 reproduces only the mass fractions' average values together with corresponding values of the UCC [17], NASC [18], PAAS [8], SMWR [19], as well as AWSS [20].

In interpreting compositional data of river sediments and adjacent soils, the geochemistry of local formations usually plays a determinant role. Additionally, due to a variable mobility, different mineral fractions are mixed and deposited along river beds depending on water speed, grain sizes and specific gravity. In this regard, Figure 2 illustrates the mass fractions of investigated elements in sediments normalized to UCC (Figure 2) [16], and in soils normalized to SMWR (Figure 2) [19]. The UCC [17] was chosen as an appropriate descriptor of the continental crust, while SMWR [19] drew on the lithogenesis theory [39], according to which, the mobilization of suspended and dissolved materials in drainage basins represent the initial stage of pedogenesis.

Here, a significant fraction of chemical elements including highly mobile alkaline and calc-alkaline ones were transported as suspended sediments and deposited along river beds and banks. Therefore, other descriptors such as NASC [17], AWSS [19] or PAAS [8] showed to be almost identical.

According to Table 1 and [36], the elemental composition of sediments and soils appears, excepting chalcophile Zn, As, Sb ad Hg, closer to UCC [17] and SMWR [19]. After a careful analysis of the experimental data presented in [36], mass fractions of potentially contaminating elements As, Sb and Hg showed increased values only for samples collected in the vicinity of mines and adits located on the Djidjikrut, Kanchoch, Chore and Mogiyon tributaries (Figures 1a, 2 and A1).

This finding is well evidenced in Figure 3a, where the As, Sb and Hg standard deviations appear disproportionately high due to the fact that in a very few places, such as mines or mine adits, their mass fractions exceeded the UCC [17] one by two to three orders of magnitude. Excepting these, the average mass fractions of investigated elements were shown to be relatively closer in both sediments and soils, as suggested by ANOVA analysis (Table 2) and by both biplots reproduced in Figure 3. In addition, a Spearman's correlation coefficient of 0.83 sustains this observation (Figure 3b).

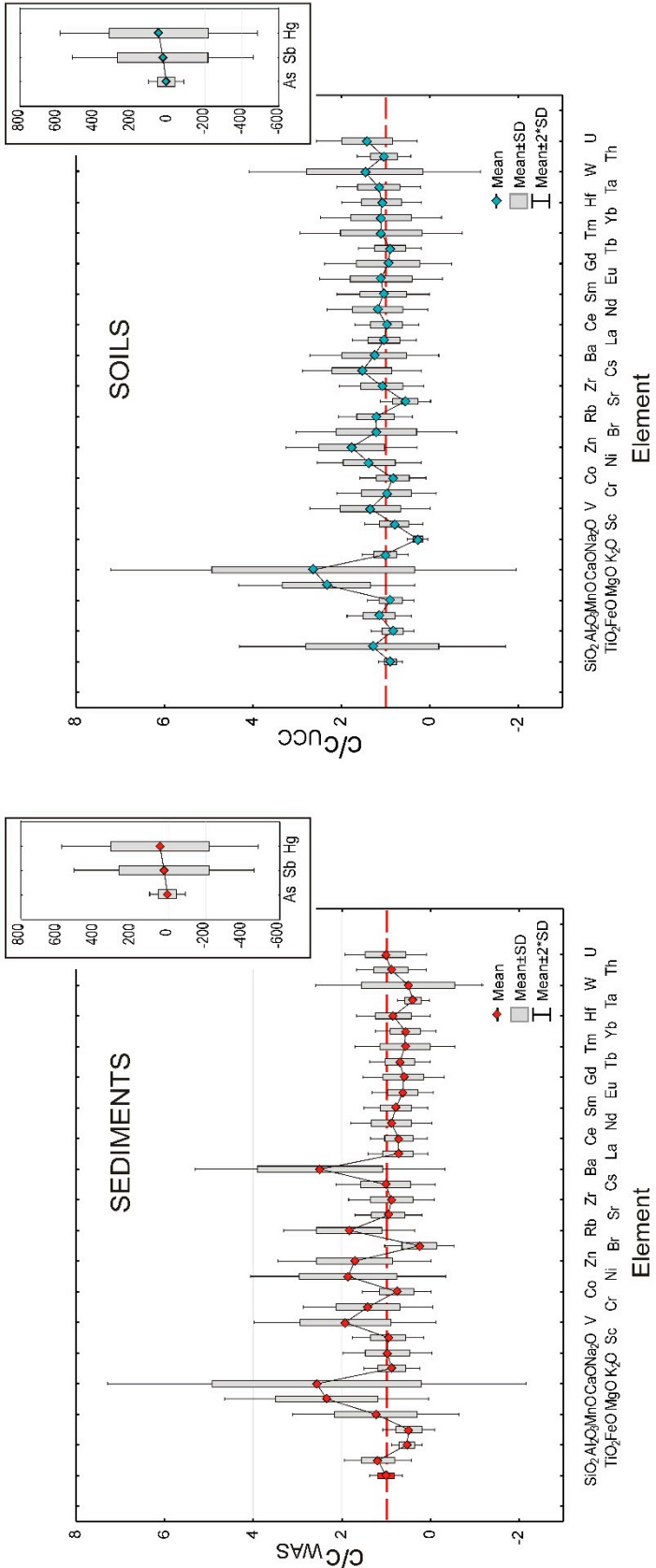

**Figure 2.** Box-plots illustrating the distribution of investigated elements in Zarafshon sediments (**lower**); and soils (**upper**). All mass fractions of soils normalized to UCC [17] and sediments to SMWR [19]. The values corresponding to As and Hg are divided by 10 and by 100 in the case of Sb.

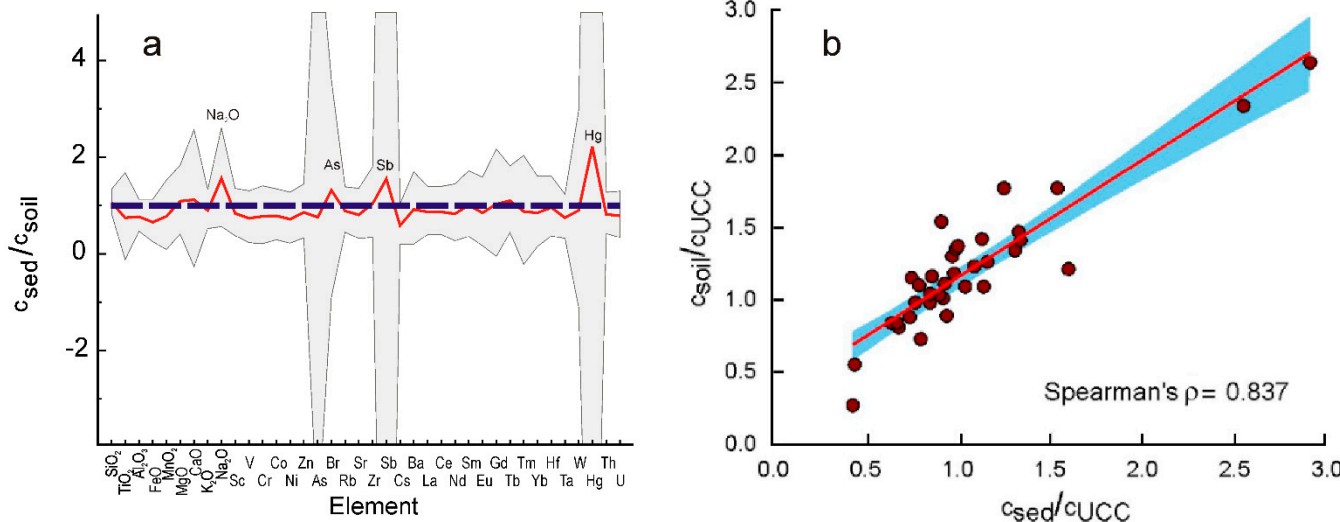

**Figure 3.** The sediment to soil ratio of mass fractions of investigated elements (red line) as well as the corresponding standard deviation (gray background) (**a**); and a biplot illustrating the reciprocal distribution of investigated elements in sediments and soils excepting the Zn, As, Sb and Hg, i.e., elements whose mass fractions in some places exceeds the UCC [17] ones by two to three orders of magnitude (**b**) (Table 1) [36]. Spearman's correlation coefficient of 0.837 at *p* < 0.01.

**Table 2.** The results of two-sample ANOVA tests illustrating the probability the major and trace elements being closer in soils and sediments. The results are based on the average mass fractions values reproduced in Table 1.

| | ANOVA Two-Sample Test | | | |
|---|---|---|---|---|
| **Elements** | **T-Test Equal Mean** | **F-Test Equal Variance** | **Mann–Whitney Equal Median** | **Anderson–Darling Equal Distribution** |
| Major elements | 0.99 | 0.87 | 1.00 | 1.00 |
| Trace elements | 0.77 | 0.59 | 0.73 | 0.97 |

Major elements: $SiO_2$, $TiO_2$, $Al_2O_3$, FeO, $MnO_2$, MgO, CaO, $K_2O$, $Na_2O$
Trace elements: Sc, V, Cr, Co, Ni, Zn, As, Br, Rb, Sr, Zr, Sb, Cs, Ba, La, Ce, Nd, Sm, Eu, Gd, Tb, Tm, Yb, Hf, Ta, W, Hg, Th, U

Figure 3a evidenced also that the ratio between the same elements in sediments and soil slightly varies around one, excepting the above mentioned PCE As, Sb and Hg and, to a lesser extent, Na of which the average mass fraction was slightly higher in sediments than in soils, as will be discussed in the next section.

### 5.1. Major Elements

The average values of the mass fractions of major, rock-forming elements excepting for MgO and CaO showed to be relatively closer to UCC [17], NASC, [18], PAAS [8], or other sedimentary systems such as SMWR [19], or Average World Suspended Sediments AWSS [20] (Table 1).

Silica represents the major constituent of all investigated samples, its mass fraction of 34.64 wt.% to 84 wt.% (62.8 ± 11.9 wt.% average) for sediments and of 31.53 wt.% to 75.12 wt.% (59.4 ± 8.8 wt.% average) for soils. On discriminating diagrams $Na_2O$ + $K_2O$ vs. $SiO_2$ (Figure 4a), $SiO_2$-$Al_2O_3$-$Na_2O$ + $K_2O$ + CaO (Figure 4b) and $K_2O$-$Al_2O_3$-$Na_2O$ + CaO (Figure 4c) the composition of investigated materials points towards a somewhat felsic to intermediate origin with a small contribution of mafic material as was the case in the Pete intrusion [5,24] (Table 1). Its average mass fraction was closer to the UCC [17] but with only a few exceptions in the sediments collected in the vicinity of the Anzob tunnel, Sarvoda 1, and Sarvoda 2 as well as soils from Chore 1 and 3, Djidjikrut 1, Obi Sara 1, Saritag 2, as well

as Sarvoda 1 to Sarvoda 7 (Figures 1a and A1). In all locations, the lower content of silica is due to an increased content of calcium oxide of which mass fractions varied between 26 and 47 wt.% (Table 1).

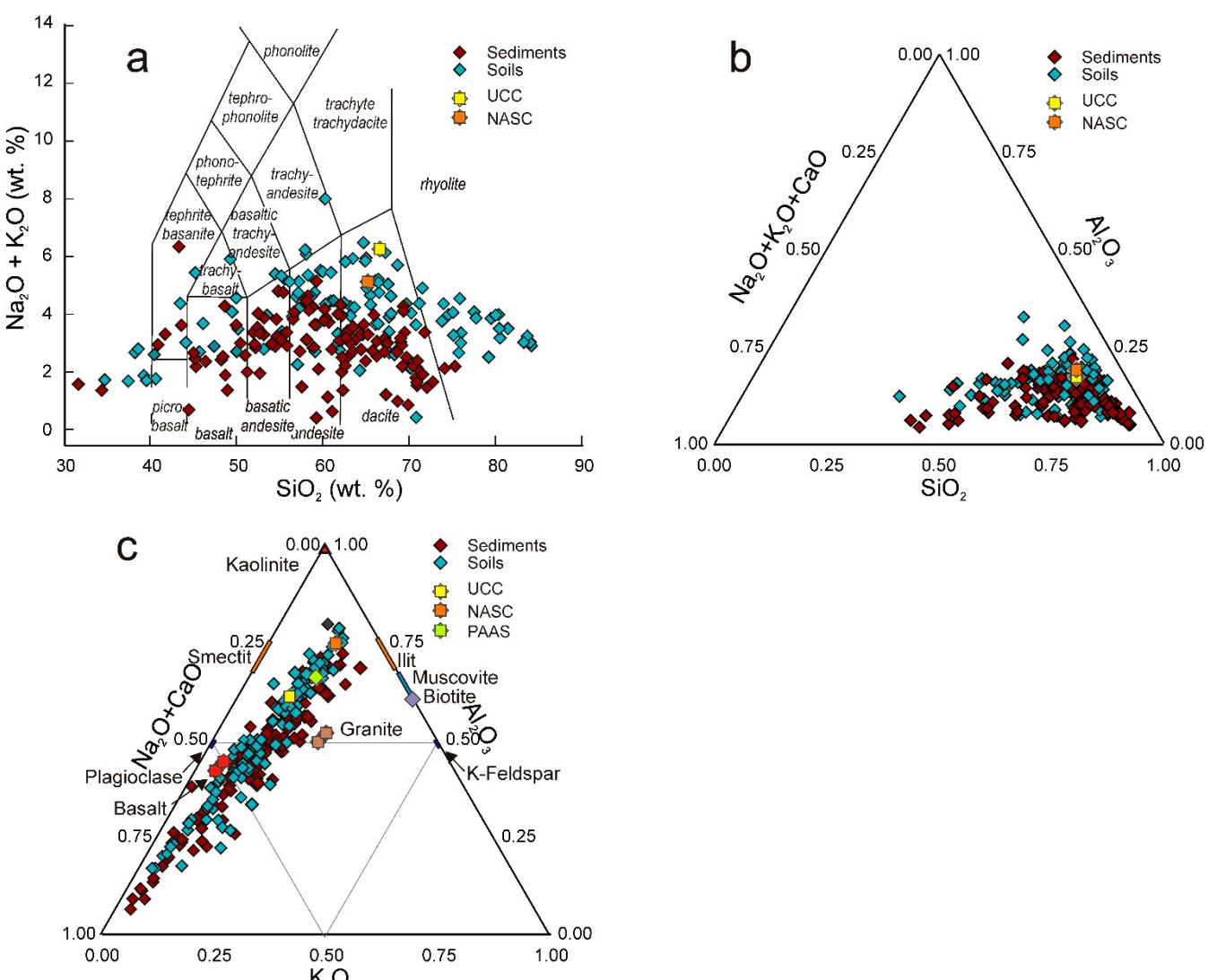

**Figure 4.** Discriminating total alkali metal oxides vs. $SiO_2$ biplot (**a**); ternary $SiO_2$-$Al_2O_3$-$Na_2O + K_2O$ + $CaO$ (**b**); and K-A-CN (**c**) diagrams illustrating the preponderance of felsic material in Zarafshon sediments and soils as well as a small fraction of weathered material.

Aluminum oxide showed in sediments an average mass fraction 9.91 ± 3.2 wt.%, smaller than the soils' one of 12.92 ± 3.7 wt.%, both of them lower than the UCC [17] and NASC [17] (Table 1). At the same time, the sodium oxide has an average mass fraction in sediments and especially in soils smaller than the corresponding UCC [17] (Table 1), suggesting a low content of plagioclase, and by consequence predominance of a sedimentary substrate (Figure 4a,b). This finding is illustrated by the discriminating diagram $SiO_2$-$Al_2O_3$-$Na_2O + K_2O$ + $CaO$ (Figure 4b) where some points are shifted towards lower $SiO_2$ content, which could be due to an increased content of CaO whose mass fraction in some places reached 28% [36]. It is worth remarking that CaO correlates positively only with MgO, in sediments as well as in soils (Table A3), suggesting the presence of dolomite $CaMg(CO_3)_2$. Besides these, the K-A-CN ternary diagram suggests a weak influence of weathering on both sediments and soils, expressed by a reduced fraction of total alkali–calc-alkali oxides $Na_2O + CaO$ of which normalized values appear smaller than 0.25 (Figure 4c).

The mass fractions of the other major elements oxides, i.e., $TiO_2$, FeO, and MnO were closer to the UCC [17] values in conformity with the previous suppositions concerning a rather felsic origin of investigated material (Table 1).

### 5.2. Trace Elements

More data concerning the origin of investigated sedimentary materials can be inferred analyzing the distribution of trace elements and especially, as mentioned in the Introduction, of those considered incompatible and insoluble such as Sc, Zr, La, Ce, Nd, Sm, Eu, Gd, Tm, Yb, Th as well as U and, to a lower extent, Cr and Ni [6,7,40].

Notably, the felsic rocks are richer in Th and light REE, whereas mafic rocks contain more Sc, Cr, Co, Ni and an increased fraction of heavy REE [41]. In this regard, Sc of average mass fraction is less than 20 mg/kg in felsic rocks but increases to 20–40 mg/kg in mafic rocks, showing it to be one of the most appropriate discriminating elements [42]. In the case of investigated sediments and soils, Sc mass fractions were $9.56 \pm 4$ and $11.4 \pm 4.6$ mg/kg, respectively, confirming the previous inference concerning the felsic origin of Zarafshon valley sedimentary material.

The same finding was confirmed by discriminating Th/Co vs. La/Sc bi-plot [43] (Figure 5a) and ternary Sc-La-Th [7,44] diagram (Figure 5b). On both diagrams, the majority of Zarafshon experimental data regarding either sediments or soils are distributed around UCC [17] and NASC [18]. Moreover, on the Th/Co vs. La/Sc bi-plot [43], all Zarafshon data are closely grouped on the felsic sector of discriminating bi-plot reproduced in Figure 5a.

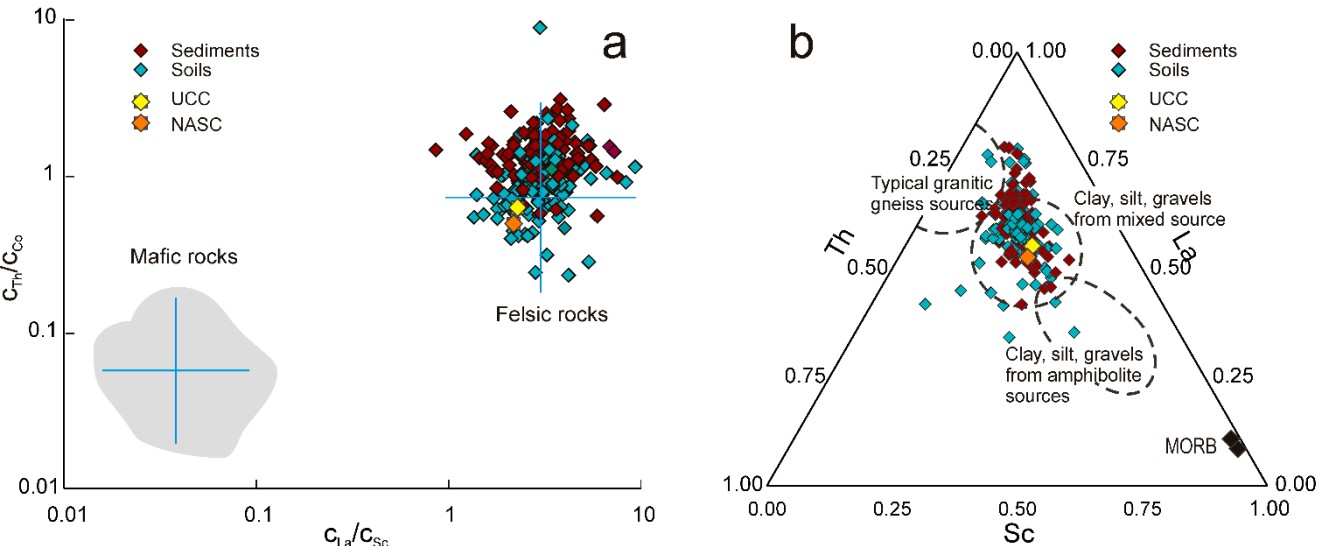

**Figure 5.** Discriminating bi-plots La/Th vs. Hf (**a**) and Th/Sc vs. Zr/Sc; (**b**) confirming the felsic origin of Zarafshon sediments and soils as well as the absence of a significant recycling of sedimentary material.

Two relatively closer bi-plots La/Th vs. Hf (Figure 6a) [45] and Th/Sc vs. Zr/Sc (Figure 6b) [8,46,47] provide a new type of information concerning not only the source material, but also the past processes that have been shaping the current position. Indeed, Hf has an increasing tendency to be enriched during the erosion of ancient (meta) sedimentary rocks. The mineral zircon, in turn, showed to be extremely resilient during recycling, so that the higher the Zr mass fraction, the larger the sedimentary material sorting and recycling [48]. In this regard, both bi-plots confirm previous inferences concerning the felsic nature of investigated sediments and soils, as well as reducing the cycle of erosion and recycling, i.e., a relatively young material. Only sediments and soils collected around the Fondarya tributary, the Kumargi Poyon, Yangob and Chore estuaries, showed a mixed felsic/basic source characterized by a higher La/Th ratio simultaneously with a reduced

Hf mass fraction. On contrary, the increased mass fraction of Hf together with reduced values of La/Th ratio suggest the presence of old sediment sources in the Mastchohi kuhi, Shahriston and some regions of Panjekent [38].

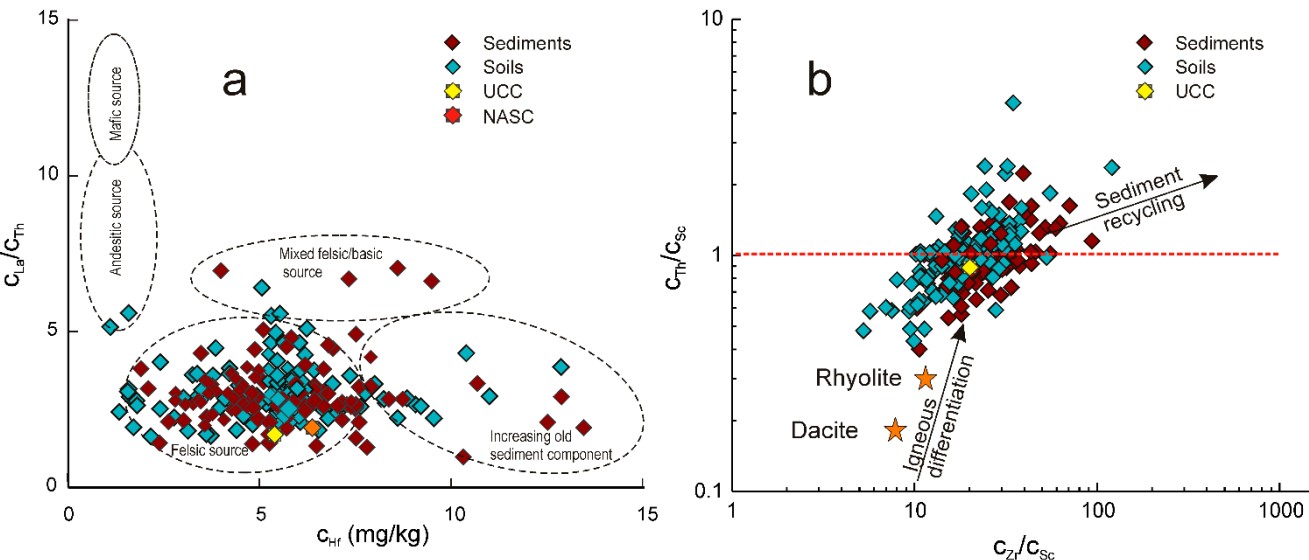

**Figure 6.** Discriminating bi-plot Th/Co vs. La/Sc (**a**); and ternary Sc-La-Th diagram (**b**) confirming the felsic origin of investigated sediments and soils.

Rare Earth Elements (REE) represent a group of 17 elements, including 15 LN, Sc and Y, all of them playing a significant role in provenance studies. Traditionally, LN are classified in light LN or light REE (LREE)—La to Sm and heavy REE (HREE)—from Eu to Lu. Since REE are not easily fractionated during sedimentation, they keep the "fingerprints" of the mother rocks making them helpful in establishing their origin [48,49]. Moreover, the Eu, which can exist in two valence states ($Eu^{2+}$ and $Eu^{3+}$) in its bivalent state due to a smaller ionic radius, can be substituted by $Sr^{2+}$ ions in plagioclase feldspars. As this process takes place only in reducing conditions which can be found only in the superior mantle or lower crust, this leads to a distinctly negative anomaly in rocks coming from the upper crust [49,50]. Another distinctive characteristic of LN, useful in evidencing the rock origin, is related to the asymmetry of LREE and HREE in felsic rocks being enriched in LREE with respect to mafic ones, depleted in this category of REE.

The La/Th ratio could be useful in inferring the provenance of sedimentary material, by considering that for the UCC [17] and NASC [18] this ratio is around 2.88 while in the case of the average Mid-Ocean Ridge Basalt (MORB) [51] it is estimated to be between 7.2 and 15.6. In the case of Zarafshon samples, La/Th ratios for sediments and soils were 3.23 ± 1 and 3.03 ± 0.9, respectively (Figure 7a), both of them closer to the UCC ratio of 2.96 [17] and of 2.53 for NASC [18]. Further analysis of the distribution function of mass fraction of eight REE, i.e., La, Ce, Nd, Sm, Eu, Gd, Tb, Tm and Yb, are shown to be closer to the UCC [16] one. This is the same peculiarity that we have noticed in the case of the La/Yb ratio, as well as in the case of Eu negative anomaly (Figure 7b, Table 3).

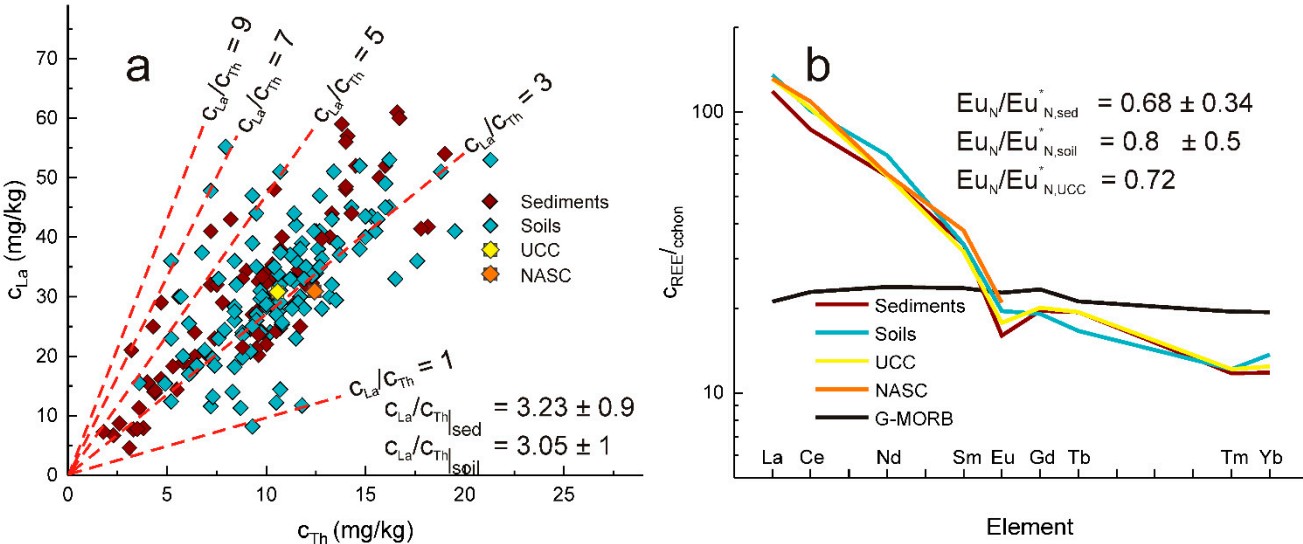

**Figure 7.** The La vs. Th bi-plot (**a**) proving that both Zarafshon sediments and soils are significantly closer to UCC [17] and NASC [18] than to MORB [51,52] as well as the eight REE average values of Zarafshon sediments and soils normalized to chondrite [53] together with the corresponding UCC data (**b**) [16].

**Table 3.** The La/Sm, Gg/Yb, Eu* as well as the Th/U values for sediments and soils. For comparison, the same ratios corresponding to UCC [17] and NASC [18] are reproduced too.

|  | La/Sm | Gd/Yb | Eu* | Th/U |
|---|---|---|---|---|
| Sediment | $4.91 \pm 1.8$ | $1.92 \pm 1.3$ | $0.69 \pm 0.27$ | $3.23 \pm 1$ |
| Soils | $5.81 \pm 2.4$ | $1.49 \pm 1.1$ | $0.8 \pm 0.32$ | $3.03 \pm 0.9$ |
| UCC | 6.6 | 1.9 | 0.72 | 2.96 |
| NASC | 5.56 | — | — | 2.53 |

A possible explanation of this small discrepancy could be related to an increased U content of which average mass fraction were between $3.04 \pm 1.38$ mg/kg in sediments and $3.8 \pm 1.5$ mg/kg in soils.

These results suggest the prevalence of the felsic material and sustains the hypotheses that sediments are the main material of the Zarafshon soils.

## 6. Conclusions

To determine at which extent the geochemistry of Turkestan and Zarafshon ranges is reflected by the adjacent sedimentary material, 116 sediments, and an equal number of soil samples were collected along the Tadjik sector of Zarafshon river and its main tributaries Arthuch, Djidjikrul, Iskadardarya, Mogiyon, Sarvoda, Shing, and Yagnob. All elemental analysis were performed by Instrumental Neutron Activation Analysis given its ability to analyze row material, without any previous treatment such as acid dissolution. This permitted determination of mass fractions of 38 major and trace elements with a sensitivity of 1 mg/kg, in both sediments and soils samples. As for each collecting point, the distance between sediments and soils sampling was no greater than 10 m, it was possible to investigate the interrelationship between sediments and soils geochemistry.

All data were compared and analyzed in correlation with similar data for the Upper Continental Crust (UCC) and North American Shale Composite (NASC). As a general conclusion, the distribution of mass fractions of all studied elements, except As, Sb and Hg showed a remarkable similarity in sediments and soils. In the case of the Presumed Contaminating Elements As, Sb and Hg, their mass fractions in few mines' adits located around Djidjikrut or Kanchoch tributaries exceeded two to three order of magnitude their

natural values. At the same time, mass fractions of both major and trace elements presented an increase variability characterized by coefficients of variation between 15 and 200% which could be attributed to the diversity of geological formation covering the Zarafshon catchment basin.

A detailed analysis of the major element oxides pointed towards a felsic origin of the collected material, with a small contribution of mafic one, as the alkali-silica discriminating diagrams $SiO_2$-$Al_2O_3$-$Na_2O$ + $K_2O$ + $CaO$ suggested. At the same time, the $K_2O$-$Al_2O_3$-$CaO$ + $Na_2O$ ternary diagram pointed towards a certain degree of weathering processes. Only in some places we have noticed an increased amount of magnesium and calcium oxides, most probable related to local deposits of limestone and dolomite. Relevant information was obtained by analyzing the distribution of low mobility and incompatible Sc, Zr, La, Ce, Nd, Sm, Eu, Gd, Tm, Yb as well as Th and U. Accordingly, Sc presented an average mass fraction of 10 to 11 mg/kg, characteristic of a felsic origin, a hypothesis suggested also by the reciprocal distribution of mass fractions of Sc, Co, La, Hf, or Th, in sediments as well as in soils. The la/Th vs. Hf and Th/Sc vs. Zr/Sc bi-plots suggest the absence of a significant sorting and recycling process of sedimentary material. The REE distribution was shown to be almost identical for sediments and soils and closer to UCC and NASC. The same pattern was noticed in the case of negative Eu anomaly, which was shown to be closer to UCC and NASC.

Given the surface of the investigated area which exceeds 17,700 km2, as well as the diversity of geological formation still insufficiently explored, the presented report should be seen as a baseline set of data for further detailed exploration.

**Author Contributions:** Conceptualization, D.A.A. and D.A.; methodology, D.A., O.G.D. and I.Z.; software, D.A. and O.G.D.; validation, O.G.D., D.A.A. and I.Z.; formal analysis, D.A. and P.S.N.; investigation, O.G.D. and D.A.; resources, D.A.A. and O.G.D.; data curation, D.A., I.Z. and O.G.D.; writing—original draft preparation, D.A.; writing—review and editing, D.A., O.G.D., I.Z. and D.A.A.; visualization, D.A. and O.G.D.; supervision, D.A.A. and O.G.D.; project administration, D.A.A.; funding acquisition, D.A.A., I.Z. and O.G.D.; All authors have read and agreed to the published version of the manuscript.

**Funding:** This research received no external funding.

**Institutional Review Board Statement:** Not applicable.

**Informed Consent Statement:** Not applicable.

**Data Availability Statement:** All experimental data concerning sampling point coordinates, mass fraction of all 38 investigated elements are accessible at: Mendeley Reference Manager as: D.A., D.A.A., O.G.D., I.Z., P.S.N. Mendeley Data, https://doi.org/10.17632/vj24br7gnt.4 (accessed on 30 July 2021).

**Acknowledgments:** The research was performed within the framework of the Cooperation Agreement between the Institute of Water Problems, Hydropower and Ecology of Academy of Sciences of Tajikistan and Sector of Neutron Activation Analysis and Applied Research of Frank Laboratory of Neutron Physics of JINR. OGD wishes to acknowledge his contribution was provided within the Cooperation Protocol no. 4920-4-20/22 between the University of Bucharest and the Joint Institute of Nuclear Research, Dubna, Russian Federation. The authors are grateful to the staff of the SNAAPI of the Frank Laboratory of Neutron Physics of JINR for carrying out the neutron activation analysis of samples. We thank two anonymous reviewers for their comments and useful suggestions.

**Conflicts of Interest:** The authors declare no conflict of interest.

## Appendix A

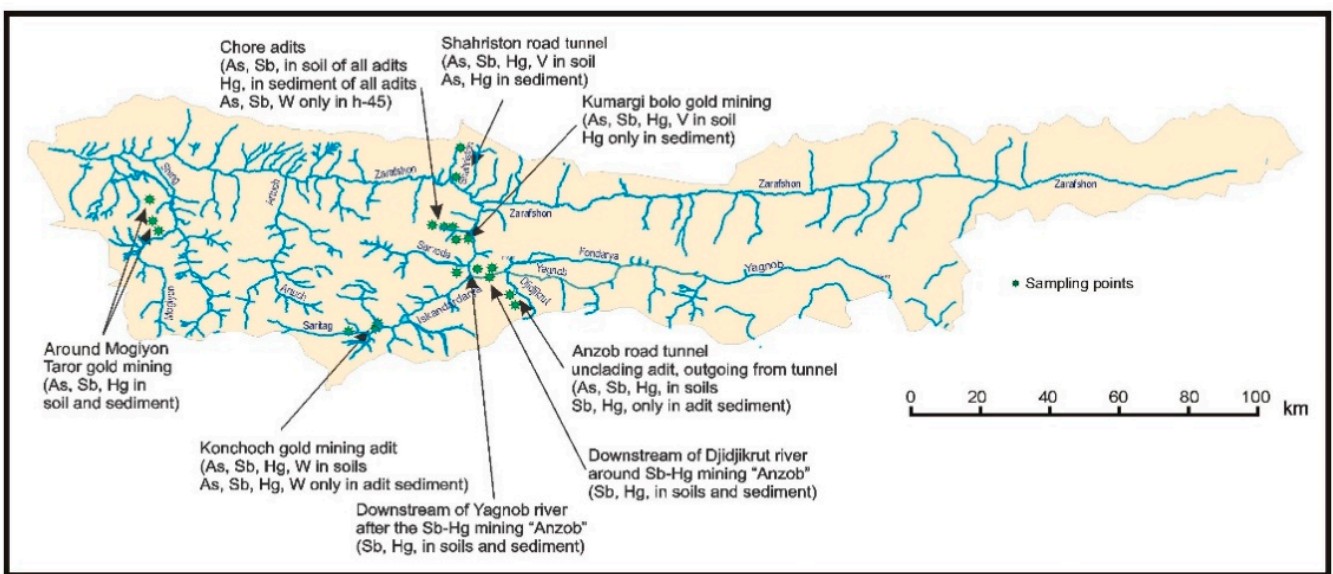

**Figure A1.** The map of Zarafshon catchment basin showing locations of the main sources of anthropogenic contamination [6].

**Table A1.** Sampling points coordinates, altitude as well as the local type of soil.

| Sampling Point | Longitude | Latitude | Altitude (m) | Soil |
|---|---|---|---|---|
| Djidjikrut 1 | 39.109142 | 68.688389 | 2726 | Regosol |
| Anzob tunnel adits | 39.108853 | 68.687314 | 2703 | Regosol |
| Djidjikrut 2 | 39.126583 | 68.682944 | 2574 | Regosol |
| Djidjikrut 3 | 39.137944 | 68.663639 | 2444 | Regosol |
| Djidjikrut 4 | 39.146806 | 68.651778 | 2398 | Regosol |
| Djidjikrut 5 | 39.164528 | 68.641861 | 2318 | Regosol |
| Djidjikrut 6 (Anzob ore-mining combine) | 39.176333 | 68.627417 | 2252 | Regosol |
| Djidjikrut 7 (Anzob ore-mining combine) | 39.199528 | 68.633278 | 1742 | Regosol |
| Yagnob 1 | 39.217333 | 69.021111 | 2351 | Regosol |
| Yagnob 2 | 39.216806 | 68.919333 | 2215 | Regosol |
| Yagnob 3 | 39.172756 | 68.874414 | 2151 | Regosol |
| Yagnob 4 (before Anzob confluence} | 39.162083 | 68.845222 | 2090 | Regosol |
| Estuary of Anzob river to the Yagnob | 39.150304 | 68.850006 | 2144 | Regosol |
| Yagnob 5 (after Anzob confluence) | 39.189667 | 68.758722 | 2070 | Regosol |
| Yagnob 6 | 39.192972 | 68.693472 | 1888 | Regosol |
| Yagnob 7 | 39.199972 | 68.640139 | 1739 | Regosol |
| Yagnob 8 | 39.194278 | 68.598278 | 1706 | Regosol |
| Yagnob 9 | 39.187472 | 68.543444 | 1657 | Regosol |
| Saritag 1 | 39.028528 | 68.271278 | 2477 | Regosol |
| Saritag 2 | 39.056417 | 68.346444 | 2194 | Regosol |

**Table A1.** *Cont.*

| Sampling Point | Longitude | Latitude | Altitude (m) | Soil |
|---|---|---|---|---|
| Konchoch | 39.042167 | 68.332750 | 2361 | Luvisol |
| Kanchoch mine adit | 39.039528 | 68.337889 | 2457 | Luvisol |
| Iskandardarya 1 | 39.101361 | 68.403583 | 2015 | Regosol |
| Iskandardarya 2 | 39.134028 | 68.453528 | 1820 | Regosol |
| Iskandardarya 3 (estuary of river) | 39.186556 | 68.534944 | 1658 | Regosol |
| Fondarya 1 | 39.196389 | 68.536028 | 1644 | Regosol |
| Alovdin Lake 1 | 39.237642 | 68.254278 | 2780 | Regosol |
| Alovdin Lake 2 | 39.237472 | 68.261083 | 2780 | Regosol |
| Sarvoda 1 | 39.250194 | 68.268889 | 2620 | Regosol |
| Sarvoda 2 | 39.262944 | 68.310917 | 2488 | Cambisol |
| Sarvoda 3 | 39.246278 | 68.355750 | 2358 | Cambisol |
| Sarvoda 4 | 39.252111 | 68.405083 | 2166 | Gleysol |
| Sarvoda 5 | 39.226611 | 68.441028 | 1958 | Gleisol |
| Sarvoda 6 | 39.223194 | 68.484611 | 1808 | Gleisol |
| Sarvoda 7—estuary of river | 39.219889 | 68.527639 | 1660 | Regosol |
| Fondarya 2 | 39.225167 | 68.530250 | 1626 | Regosol |
| Pete 1 (right tributary to Fondarya) | 39.248639 | 68.561778 | 1895 | Gleisol |
| Pete 2 (right tributary to Fondarya) | 39.249361 | 68.549889 | 1710 | Gleisol |
| Fondarya 3 | 39.258806 | 68.544000 | 1589 | Regosol |
| Kumargi Bolo 1 | 39.276861 | 68.573056 | 2029 | Regosol |
| Kumargi Bolo 2 | 39.275444 | 68.547583 | 1690 | Regosol |
| Chore 1—upper of river | 39.313639 | 68.479694 | 2148 | Regosol |
| Chore 2—adit | 39.308944 | 68.506194 | 1863 | Regosol |
| Chore 3—adit | 39.308639 | 68.514056 | 1753 | Regosol |
| Chore 4—adit | 39.308222 | 68.528694 | 1577 | Regosol |
| Chore 5—estuary of river | 39.306083 | 68.532917 | 1580 | Regosol |
| Fondarya 4 | 39.305528 | 68.533861 | 1580 | Cambisol |
| Fondarya 5 | 39.344750 | 68.554333 | 1466 | Cambisol |
| Kumargi poyon 1 | 39.325944 | 68.628389 | 3629 | Gleysol |
| Kumargi poyon 2 | 39.363167 | 68.602694 | 1782 | Gleysol |
| Kumargi poyon 3 | 39.363967 | 68.561083 | 1434 | Gleysol |
| Fondarya 6 | 39.380528 | 68.549556 | 1389 | Cambisol |
| Zarafshon 1 | 39.452667 | 70.426100 | 2732 | Regosol |
| Zarafshon 2 (Dihisor) | 39.426194 | 70.307583 | 2612 | Regosol |
| Zarafshon 3 (Dihisor) | 39.444472 | 70.189750 | 2517 | Regosol |
| Zarafshon 4 (Paldorak) | 39.457583 | 70.076194 | 2423 | Regosol |
| Zarafshon 5 (Paldorak) | 39.426417 | 69.963444 | 2330 | Regosol |
| Zarafshon 6 (Sabak) | 39.447639 | 69.841194 | 2272 | Regosol |
| Zarafshon 7 (Madrushkat) | 39.453167 | 69.723861 | 2267 | Regosol |
| Rovut– tributary to Zarafshon | 39.449194 | 69.676306 | 2242 | Regosol |
| Zarafshon 8 (Madrushkat) | 39.440358 | 69.601324 | 2158 | Regosol |
| Madrushkat– tributary to Zarafshon | 39.446944 | 69.563472 | 2123 | Regosol |

**Table A1.** *Cont.*

| Sampling Point | Longitude | Latitude | Altitude (m) | Soil |
|---|---|---|---|---|
| Tributary 2—to Zarafshon (Isizi poyon) | 39.439889 | 69.542361 | 2075 | Regosol |
| Zarafshon 9 Isizi poyon) | 39.431222 | 69.483639 | 1999 | Regosol |
| Tributary 3—to Zarafshon (Isizi poyon) | 39.415750 | 69.418694 | 1985 | Regosol |
| Zarafshon 10 (Isizi poyon) | 39.413306 | 69.376833 | 1927 | Regosol |
| Zarafshon 11 (Pastigov) | 39.413833 | 69.269194 | 1919 | Regosol |
| Kallakhona–tributary to Zarafshon | 39.418194 | 69.264861 | 1913 | Regosol |
| Oburdon (Pastigov) | 39.425611 | 69.167250 | 1875 | Regosol |
| Zarafshon 12 (Pastigov) | 39.417056 | 69.156833 | 1820 | Regosol |
| Zarafshon 13 (Shamtuch) | 39.401222 | 69.060500 | 1742 | Regosol |
| Zarafshon 14 (Veshab) | 39.402833 | 68.954028 | 1660 | Regosol |
| Zarafshon 15 (Shavadki poyon) | 39.391472 | 68.853306 | 1619 | Regosol |
| Zarafshon 16 (Rarz) | 39.380639 | 68.773361 | 1508 | Regosol |
| Zarafshon 17 (Fatmev) | 39.385861 | 68.673861 | 1462 | Regosol |
| Zarafshon 18 (Sangiston) | 39.387250 | 68.570250 | 1405 | Regosol |
| Shahriston 1, tributary right site of tunnel | 39.520806 | 68.553389 | 2728 | Regosol |
| Shahriston 2, tributary left site of tunnel | 39.519972 | 68.554889 | 2744 | Regosol |
| Shahriston 3–estuary | 39.440583 | 68.541389 | 1682 | Luvisol |
| Zarafshon 19 | 39.419455 | 68.509311 | 1366 | Regosol |
| Zarafshon 20 | 39.439194 | 68.441583 | 1306 | Regosol |
| Zarafshon 21 | 39.439194 | 68.441583 | 1284 | Regosol |
| Zarafshon 22 | 39.440806 | 68.273250 | 1226 | Regosol |
| Zarafshon 23 | 39.443472 | 68.087750 | 1206 | Regosol |
| Artuch 1 (Alpine camp) | 39.274889 | 68.139167 | 2142 | Luvisol |
| Artuch 2 | 39.335278 | 68.087750 | 1573 | Luvisol |
| Artuch 3 | 39.397806 | 68.043250 | 1267 | Regosol |
| Zarafshon 24 | 39.471667 | 67.991833 | 1113 | Gleysol |
| Zarafshon 25 | 39.466472 | 67.887139 | 1072 | Gleysol |
| Obi Sara 1 | 39.379278 | 67.880028 | 1775 | Regosol |
| Obi Sara 2 | 39.415889 | 67.865639 | 1315 | Regosol |
| Obi Sara 3 | 39.468389 | 67.872833 | 1079 | Regosol |
| Zarafshon 26 | 39.497306 | 67.785056 | 1035 | Gleysol |
| Lake 1 (Khazorchashma) | 39.113250 | 67.855250 | 2393 | Regosol |
| Lake 2 (Marguzor) | 39.140972 | 67.859528 | 2106 | Regosol |
| Lake 3 | 39.166083 | 67.838972 | 1888 | Regosol |
| Lake 4 | 39.197278 | 67.819333 | 1811 | Regosol |
| Lake 5 | 39.203564 | 67.810650 | 1718 | Regosol |
| Lake 6 | 39.210056 | 67.805028 | 1701 | Regosol |
| Lake 7 | 39.220306 | 67.800806 | 1610 | Regosol |
| Shing 1 | 39.240806 | 67.798028 | 1565 | Regosol |
| Shing 2 | 39.273667 | 67.802778 | 1382 | Regosol |
| Shing 3 industrial zone | 39.306917 | 67.783778 | 1282 | Regosol |
| Mogiyon 1 | 39.112611 | 67.733639 | 2383 | Gleysol |
| Mogiyon 2 | 39.147750 | 67.737000 | 2144 | Gleysol |
| Mogiyon 3 | 39.215194 | 67.711944 | 1722 | Gleysol |
| Mogiyon 4 | 39.273722 | 67.688639 | 1474 | Gleysol |
| Mogiyon 5 | 39.315583 | 67.770167 | 1250 | Luviysol |

**Table A1.** *Cont.*

| Sampling Point | Longitude | Latitude | Altitude (m) | Soil |
|---|---|---|---|---|
| Mogiyon 6 after junction with Shing river | 39.339583 | 67.770306 | 1222 | Regosol |
| Mogiyon 7 | 39.378778 | 67.759472 | 1180 | Regosol |
| Mogiyon 8 | 39.419444 | 67.766083 | 1099 | Regosol |
| Mogiyon 9 | 39.456556 | 67.735889 | 1041 | Regosol |
| Mogiyon 10 | 39.495083 | 67.711861 | 994 | Regosol |
| Zarafshon 27 | 39.502889 | 67.678861 | 981 | Regosol |
| Zarafshon 28 | 39.496728 | 67.552247 | 934 | Gleysol |
| Zarafshon 29 | 39.529500 | 67.433860 | 883 | Gleysol |

**Table A2.** The certified as well as experimentally determined mass fractions of the NIST 2709 CRM elements.

| Element | Certified (w%) | Measured (w%) | Element | Certified (mg/kg) | Measured (mg/kg) |
|---|---|---|---|---|---|
| Al | 7.37 | $7.11 \pm 0.16$ | Sb | 1.55 | $1.6 \pm 0.06$ |
| Ca | 1.91 | $1.99 \pm 0.09$ | Ba | 979 | $951 \pm 29$ |
| Fe | 3.36 | $3.45 \pm 0.07$ | Cr | 130 | $121 \pm 9.2$ |
| Mg | 1.46 | $1.38 \pm 0.03$ | Co | 12.8 | $13.6 \pm 0.3$ |
| K | 2.11 | $2.15 \pm 0.06$ | Mn | 529 | $546 \pm 19$ |
| Si | 30.3 | $30.95 \pm 0.41$ | Sr | 239 | $230 \pm 8$ |
| Na | 1.22 | $1.26 \pm 0.03$ | V | 110 | $121 \pm 11$ |
| Ti | 0.34 | $0.3 \pm 0.01$ | Zr | 195 | $201 \pm 50$ |

**Table A3.** The matrix of Spearman's correlation coefficients of major elements (as oxides) in soils and sediments. All correlations significant at $p < 0.05$ are represented in red (positive correlation) or blue (negative correlation) color.

| Sediments | $SiO_2$ | $TiO_2$ | $Al_2O_3$ | FeO | $MnO_2$ | MgO | CaO | $K_2O$ |
|---|---|---|---|---|---|---|---|---|
| $TiO_2$ | −0.251 | | | | | | | |
| $Al_2O_3$ | −0.302 | **0.840** | | | | | | |
| FeO | −0.340 | **0.758** | **0.819** | | | | | |
| $MnO_2$ | −0.018 | **0.621** | **0.639** | **0.510** | | | | |
| MgO | **−0.804** | 0.381 | 0.357 | 0.382 | 0.013 | | | |
| CaO | **−0.752** | −0.195 | −0.184 | −0.153 | −0.191 | **0.577** | | |
| $K_2O$ | −0.155 | **0.612** | **0.627** | **0.728** | 0.392 | 0.268 | −0.260 | |
| $Na_2O$ | 0.204 | 0.419 | 0.410 | 0.276 | 0.313 | −0.066 | −0.435 | 0.262 |
| **Soils** | $SiO_2$ | $TiO_2$ | $Al_2O_3$ | FeO | $MnO_2$ | MgO | CaO | $K_2O$ |
| $TiO_2$ | −0.081 | | | | | | | |
| $Al_2O_3$ | −0.119 | **0.645** | | | | | | |
| FeO | −0.291 | 0.445 | 0.348 | | | | | |
| $MnO_2$ | −0.009 | 0.442 | 0.456 | 0.300 | | | | |
| MgO | **−0.671** | 0.062 | −0.035 | 0.153 | 0.022 | | | |
| CaO | **−0.668** | −0.436 | **−0.527** | −0.123 | −0.356 | 0.463 | | |
| $K_2O$ | −0.213 | **0.543** | **0.626** | 0.459 | 0.255 | 0.108 | −0.276 | |
| $Na_2O$ | −0.081 | **0.536** | 0.403 | 0.422 | 0.378 | −0.011 | −0.268 | 0.431 |

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
