# Peer review of "On the Geochemistry of Major and Trace Elements Distribution in Sediments and Soils of Zarafshon River Valley, Western Tajikistan"

_applsci, doi:10.3390/app12062763_

Round 1
Reviewer 1 Report
The manuscript on the distribution of major and trace elements in sediments and soils of the Zarafshon River Valley by Djamshed and others includes considerable analytical data and represents a serious effort. The objectives were clearly stated as 1) to determine the extent to which the sediment and soil samples collected along the river valley "... can be related to crustal materials ..." (line 95) and 2) to determine "... any interrelationship between sediments and adjacent soils" (line 95). Several samples of soil and sediment were collected and carefully analyzed for both major and trace elements. However, it is never made clear what their definition of soil is and how it differs from sediment. Most soil is layered into a, b, and c horizons and clearly defined on the basis of clay, silt and sand content among other characteristics. So, soil scientists will be disappointed by the lack of details pertaining to the soils. Then in the methods sections the authors point out that the soil sample columns were collected no longer than 10 cm (line 160). Such thin soil sections are poorly developed, but it needs to be made clear what criteria was used to distinguish soil from sediment. Is "soil" anything less than 10 cm deep? The authors describe the chemical differences between the soil samples and the sediment in great detail but again, most soil scientists will be looking for exchange reactions and the typical soil descriptions. Which elements were leached out and what is the clay content? What are the differences in the up-stream soil compared with the down-stream soil or are there absolutely no differences? These kinds of descriptive details are absent from the study and fail to address the second objective (interrelationship between sediment and soil). What are the differences in the up-stream soil compared with the down-stream soil? The first objective (relationship to crustal material) is also a bit inconclusive. It is difficult to determine the relationship of the soil and sediment to the bedrock without knowing anything about the geology of the bedrock. Therefore, a geologic map is almost necessary but is not provided. Abundant evidence is provided estimating the provonance, or lithology of the source, but it would be much easier to map out the closest bedrock outcrops uphill from the river valley or to consult some kind of geologic map. If that can't be done the reason why should be given.
A few less important problems:
line 245. What do the authors intend by "in the vicinity of mines and adits"? How may meters down-stream from the mines are high values of As, Sb, and Hg found and how high do the values get?
line 353. Always define the authors of the diagrams you use in the figure captions.
line 394. The authors state that the chemistry of the sediments and the soils is "remarkably similar except As, Sb, and Hg". This is important and should be explained in some detail.
Each of the items that I have suggested could easily be done and would improve the revised draft that I would encourage you to submit.
Author Response
Dear Sir,
Thank you for your remarks and suggestions which we had tried to take into account in the new, revised version of our manuscript.
Below I will present your remarks and our answers.
With my best wishes,
Octavian Duliu
The manuscript on the distribution of major and trace elements in sediments and soils of the Zarafshon River Valley by Djamshed and others includes considerable analytical data and represents a serious effort. The objectives were clearly stated as 1) to determine the extent to which the sediment and soil samples collected along the river valley "... can be related to crustal materials ..." (line 95) and 2) to determine "... any interrelationship between sediments and adjacent soils" (line 95). Several samples of soil and sediment were collected and carefully analyzed for both major and trace elements.
Remark
However, it is never made clear what their definition of soil is and how it differs from sediment. Most soil is layered into a, b, and c horizons and clearly defined on the basis of clay, silt and sand content among other characteristics. So, soil scientists will be disappointed by the lack of details pertaining to the soils.
Answer:
We have mentioned: Introduction - Rows: 78-87
According to [14] soil is a result of the combined action of weather, geographic relief, organisms and human activity on parent mineral substrate, all of them continuously interacting. The actual soil is no older than Pleistocene, i.e 2.59 My, almost coincident with the onset of Northern hemisphere glaciation.
At its turn, sediments represent a fragmented mineral substrate under the action of weathering and erosion and which is transported at long distances by the action of water, wind or ice and finally deposited. With respect to soil, sediments consist of unconsolidated clastic material with a reduced amount of organic matter. When consolidated by natural cementation processes, sediments turn into sedimentary rocks such as siltstone, sandstone, shale, etc.
Both sediments and soils preserve the fingerprint of parental mineral substrate, which, under the action of external factors, evolve in different manner. Collected from the same places and correlated investigated, these sedimentary materials could bring a plus of information concerning the local geochemistry as well as their interrelationship.”
Remark:
Then in the methods sections the authors point out that the soil sample columns were collected no longer than 10 cm (line 160). Such thin soil sections are poorly developed, but it needs to be made clear what criteria was used to distinguish soil from sediment. Is "soil" anything less than 10 cm deep?
Answer:
We have explained: Materials and Methods Rows: 319-321
As at altitudes greater than 2000 m the soil cover is less developed, soil columns were no longer than 10 cm. For our study we have retained the top soil (horizon A), as we were interested also to evidence recent contamination processes [6].
Remark:
The authors describe the chemical differences between the soil samples and the sediment in great detail but again, most soil scientists will be looking for exchange reactions and the typical soil descriptions.
Answer:
The surface of Zarafshon catchment basin is of 17 7000 km2, so to describe in details different type of soils would take a lot of space, but we have included in the Appendix table A1 which presents in details for each sampling point the main type of soil.
Remark:
Which elements were leached out and what is the clay content? What are the differences in the up-stream soil compared with the down-stream soil or are there absolutely no differences? These kinds of descriptive details are absent from the study and fail to address the second objective (interrelationship between sediment and soil).
Answer:
To answer this remark, we have calculated and represented in the newly included Fig 3a the ratio between average mass fractions of all elements in sediments and soils followed by a short discussion of these findings.
Rows 403-408
According to Table 1 and [36], the elemental composition of sediments and soils appears, excepting chalcophile Zn, As, Sb ad Hg, closer to UCC [17] and SMWR [19]. At a careful analysis of the experimental data presented in [36], mass fractions of potentially contaminating elements As, Sb and Hg showed increased values only for samples collected in the vicinity of mines and adits located on Djidjikrut, Kanchoch, Chore and Mogiyon tributaries (Fig. 1a, 2a, A1).
Rows 427 – 435
This finding is well evidenced on Fig. 3a, where the As, Sb and Hg standar deviations apear dispopotionately high. Excepting these, the average mass fractions of investigated elements were shown to be relatively closer in both sediments and soils, as suggested by ANOVA analysis (Table 2) and by both biplots reproduced in Fig. 3. Also, a Spearman’s correlation coefficient of 0.83 susteins this observation (Fig. 3b).
Figure 3a evidenced also that the ratio between the same elements in sediments and soil slightly varies around one, excepting above mentioned PCE As, Sb and Hg and at a lesser extent Na of which average mass fraction was slightly higher in sediments than in soils as will be discussed in the next section.
Remark
What are the differences in the up-stream soil compared with the down-stream soil? The first objective (relationship to crustal material) is also a bit inconclusive. It is difficult to determine the relationship of the soil and sediment to the bedrock without knowing anything about the geology of the bedrock.
Therefore, a geologic map is almost necessary but is not provided. Abundant evidence is provided estimating the provenance, or lithology of the source, but it would be much easier to map out the closest bedrock outcrops uphill from the river valley or to consult some kind of geologic map. If that can't be done the reason why should be given.
Answer:
We totally agree with this remark, but the only geological map of Tajikistan is of a poor resolution e.g.150 dpi which make it impossible not only to reproduce but also to decipher it. Therefore, to compensate for this weakness, we have included more descriptions of the geology of investigated area: see section 3 Geological setting
A few less important problems:
Remark:
line 245. What do the authors intend by "in the vicinity of mines and adits"? How may meters down-stream from the mines are high values of As, Sb, and Hg found and how high do the values get?
Answer:
We have mentioned Materials and Methods – Sampling Lines 330-334
To evidence any correlation between sediments and soils geochemistry, with two exceptions, in all other cases, sediment and soil samples were collected in the same places, the distance between the sediments and the soils sampling location never exceeding 5-10 m (Table 1) and [36]. The same distances separated mine, adits or tunnels of the downstream collecting places”.
Remark:
Always define the authors of the diagrams you use in the figure captions.
Answer:
It is a quite unusual remark which. I have never seen on other papers. For your information Dr. Amadzoda and me are the sole authors of all figures. Partially we have answer to this remark as follows: visualization, D.A., and O.G.D.
Remark: line 394.
The authors state that the chemistry of the sediments and the soils is "remarkably similar except As, Sb, and Hg". This is important and should be explained in some detail.
Answer
We have mentioned three times:
Results and Discussion lines: 423-425
…. and soils excepting the Zn, As, Sb and Hg, i.e., elements whose mass fractions in some places exceeds the UCC [17] ones by two to three orders of magnitude (b) (Table 1) [36].
Lines: 427 - 430
This finding is well evidenced on Fig. 3a, where the As, Sb and Hg standard deviations appear disproportionately high due to the fact that in very few places, such as mines of mine adits their mass fractions exceeded two to three order of magnitude the UCC [17] one. Excepting these, the average mass fractions of investigated elements were shown to be relatively closer in both sediments and soils, as suggested by ANOVA analysis (Table 2) and by both biplots reproduced in Fig. 3. Also, a Spearman’s correlation coefficient of 0.83 sustains this observation (Fig. 3b).
Figure 3a evidenced also that the ratio between the same elements in sediments and soil slightly varies around one, excepting above mentioned PCE As, Sb and Hg and at a lesser extent Na of which average mass fraction was slightly higher in sediments than in soils as will be discussed in the next section.
Conclusions lines: 579 – 582
In the case of Presumably Contaminating Elements As, Sb and Hg, their presence in few mines adits located around Djidjikrut or Kanchoch tributaries where their mass fractions exceeded two to three order of magnitude their natural values.
Each of the items that I have suggested could easily be done and would improve the revised draft that I would encourage you to submit
Answer
Once more we thank you for help in increasing manuscript quality

Reviewer 2 Report
This article has many positive attributes, and in my judgement applied sciences is the appropriate journal for this article.
Before I accept the article, I would like for the authors to address the following points.
I would like the authors to expand a little more on the introduction, as the analysis of trace elements in sediments is of upmost importance. I believe the following article would strengthen the introduction, and even help get this article more citations. The following article details the analysis of sediments for trace elements in a polluted area.
Bussan, Derek, Austin Harris, and Chris Douvris. "Monitoring of selected trace elements in sediments of heavily industrialized areas in Calcasieu Parish, Louisiana, United States by inductively coupled plasma-optical emission spectroscopy (ICP-OES)." Microchemical Journal 144 (2019): 51-55.
Line 167: What type of acidified water was used? Just as an example a reader could interpret acidified water as water with vinegar in it. Also, what type of water was used, was it deionized, distilled, tap water. This is important to identify, as different water sources could be more prone to contamination.
Just out of curiosity, I am wondering why the authors analyzed CRM 1547 Peach Leaves, as this is a different matrix than soils or sediments.
For the percent recoveries of the reference materials this manuscript could be strengthened if the authors could add a table showing the percent recoveries of each element with each reference material. This would aid the reader regarding the accuracy of this technique. As the reference materials have different elemental concentrations as well as to compare the different matrices.
Overall, there are a sufficient number of sample and the article is well written. I would accept the article with a few minor revisions after the authors addressed my suggestions and questions listed above.
Author Response
Dear Sir,
Firs of all we thank you for the remarks and suggestions to which we have tried answer in a proper manner.
Below we have enumerated your remakes and our corresponding answers.
With our best wishes,
Octavian Duliu
This article has many positive attributes, and in my judgement applied sciences is the appropriate journal for this article.
Before I accept the article, I would like for the authors to address the following points.
Remark
I would like the authors to expand a little more on the introduction, as the analysis of trace elements in sediments is of upmost importance. I believe the following article would strengthen the introduction, and even help get this article more citations. The following article details the analysis of sediments for trace elements in a polluted area.
Bussan, Derek, Austin Harris, and Chris Douvris. "Monitoring of selected trace elements in sediments of heavily industrialized areas in Calcasieu Parish, Louisiana, United States by inductively coupled plasma-optical emission spectroscopy (ICP-OES)." Microchemical Journal 144 (2019): 51-55.
Answer
We have included in the Introduction more refences to trace elements Lines 60-63
Although the total population of Zarafshon valley does not exceed 400,000 inhabitants, mainly populating small villages, the existence of non-ferrous ore outcrops could represent the main source of down-stream contamination, previously analyzed in [6].
And Lines 70-87
Besides them, there are more other trace elements such as V, Cr, Co, Ni, Zn, As, Sb, or Hg, intensively used in diverse industrial processes, and, of which presence at levels significantly exceeding the natural ones are related to an anthropogenic contamination which could be industrial [11,12] or due to urban as well as rural agglomerations [13]. According to [14] soil is a result of the combined action of weather, geographic relief, organisms and human activity on parent mineral substrate, all of them continuously interacting. The actual soil is no older than Pleistocene, i.e 2.59 My, almost coincident with the onset of Northern hemisphere glaciation.
At its turn, sediments represent a fragmented mineral substrate under the action of weathering and erosion and which is transported at long distances by the action of water, wind or ice and finally deposited. With respect to soil, sediments consist of unconsolidated clastic material with a reduced amount of organic matter. When consolidated by natural cementation processes, sediments turn into sedimentary rocks such as siltstone, sandstone, shale, etc.
Both sediments and soils preserve the fingerprint of parental mineral substrate, which, under the action of external factors, evolve in different manner. Collected from the same places and correlated investigated, these sedimentary materials could bring a plus of information concerning the local geochemistry as well as their interrelationship.
As well as Ref. 11-14
11. Bussan, D., Harris A., Douvris C. Monitoring of selected trace elements in sediments of heavily industrialized areas in Calcasieu Parish, Louisiana, United States by inductively coupled plasma-optical emission spectroscopy (ICP-OES), Microchem. J. 2019, 144, 51-55, https://doi.org/10.1016/j.microc.2018.08.053
12. Belozertseva I.A., Milić M., Tošić S., Saljnikov E. Environmental Pollution in the Vicinity of an Aluminium Smelter in Siberia. In: Saljnikov E., Mueller L., Lavrishchev A., Eulenstein F. (eds) Advances in Understanding Soil Degradation. Innovations in Landscape Research. Springer, 2022, pp. 379-402, https://doi.org/10.1007/978-3-030-85682-3_18
13. Badawy W.M., Sarhan Y., Duliu O.G., Kim J., El Samman H., Hussein A.A., Frontasyeva M., Shcheglov A. Monitoring of air pollutants using plants and co-located soil—Egypt: characteristics, pollution, and toxicity impact. Environ. Sci. Pollut. Res. 2021, https://doi.org/10.1007/s11356-021-17218-7
14. Ponge, J.-F. The soil as an ecosystem, Bio. Fertil. Soils. 2015, 51, 645–648, https://doi.org/10.1007/s00374-015-1016-1. S2CID 18251180.
Remark:
Line 167: What type of acidified water was used? Just as an example a reader could interpret acidified water as water with vinegar in it. Also, what type of water was used, was it deionized, distilled, tap water. This is important to identify, as different water sources could be more prone to contamination.
Answer
We have explained Lines: 327-329
Any possible cross-contamination was avoided by washing the glass sampler before any sampling with a solution of diluted hydrochloric acid with deionized water and soaking them with disposable cotton serves.
Remark
Just out of curiosity, I am wondering why the authors analyzed CRM 1547 Peach Leaves, as this is a different matrix than soils or sediments.
Answer
The CRM 1547 Peache Leaves is a part of GSS, Lines: 370-384
Further, by using all the above-mentioned SRM it was realized a Group of Standard Sample (GSS) [37,38] which, permitted to choose the most appropriate SRM lines to minimize the error in determining the mass fractions of all investigated elements. Regarding it, is worth mentioning that GSS is a proprietary software which, besides helping to determine the mass fractions of chosen elements, it allowed calculating to which extent the mass fractions of SRM corresponds to certified ones [37,38]. Following this procedure, the accuracy quantified by means of standard deviation corresponding to each group of three aliquots was lower than 10%. In the case of SRM, the CSU varied between 3 and 15%, higher in the case of REE, but never greater than 20%. It is worth mentioning that for all 38 investigated elements, the detection limit was of 1 mg/kg and lower [37,38].
To illustrate the sensitivity and accuracy of INAA measurements following the CSS software, in Table A2 we have reproduced, as an example, the experimentally measured mass fractions of SRM 2709. Similar determinations were done in the case of all utilized SRM.
Remark
For the percent recoveries of the reference materials this manuscript could be strengthened if the authors could add a table showing the percent recoveries of each element with each reference material. This would aid the reader regarding the accuracy of this technique. As the reference materials have different elemental concentrations as well as to compare the different matrices.
Answer
We have included the table A2 (Appendix)
Overall, there are a sufficient number of sample and the article is well written. I would accept the article with a few minor revisions after the authors addressed my suggestions and questions listed above.
Once more thank you for all remarks and suggestions.

This manuscript is a resubmission of an earlier submission. The following is a list of the peer review reports and author responses from that submission.
Round 1
Reviewer 1 Report
The manuscript needs significant improvement in the English language presentation. Figure 1 needs to be in the Introduction, even though 1b shows the sample locations. Expanding the size of some figures will allow better resolution for the many data points.
This manuscript should be revised and submitted for a re-review.
Author Response
We have thoroughly revised the manuscript, in some places removing the non-essential information such as the chemical formula of some minerals.
We have increased the dimension including the lettering. Now it occupies an entire page.
We have expanded all figures to fill all page.

Reviewer 2 Report
There are several serious problems with the manuscript "Geochemical Features of Major and Trace Element ...." by Abdushukurov and others before it will be ready for publication. Major problems include:
- Beginning with the title and abstract; since few people will know the location of the Zarafshon River Valley the title should include the country "Tajikistan" at the end of the title and at the end of the first sentence of the abstract.
- The abstract and most of the text need to be carefully edited for English grammar. Almost every sentence is in need of revision.
- Exactly how were the soil samples distinguished from the sediment samples? At what depth did you collect the samples; how thick were the soil horizons; how much organic content was in the soil before it was chemically analyzed; how did you find soil samples on bedrock, desert sand surfaces, or mine drifts; were all the sediment samples unconsolidated? What kind of soils did you sample; very important.
- The scale of most of the figures, particularly Fig. 1 is way too small. None of the detail described in the text is readable in the figures.
- The fact that the sediment and soil samples were collected from the same locations is not made clear until Line 322. This fact should be in your abstract and emphasized early in the text. Unless they were collected from the same locations the whole manuscript would be meaningless and there would be no way to compare the soils with the sediments. The fact that they were gives some credibility to the project and actually makes the manuscript interesting.
- An important section of the manuscript that should be developed is a chemical comparison of the soils using Table 1. Which elements were leached out of the soils compared to the underlying sediments and which elements tended to build up in the soils as insoluble residues (such as aluminum) assuming they were residual soils.?
- Most of the second half of the manuscript is devoted to the provenance of the sediments. This is not a very interesting question and could be easily estimated with even the most basic geologic map. The absence of any geologic map is a major problem. Surely there must be at least some kind of geologic map even if it was published a long time ago. With all the mining activity in the valley I would suspect at least something.
Other needed revisions:
Line 49. Define sierozem.
Line 139. How many samples of paralavas?
Line 157. How many samples of the upper flows of the Iskandardarya River? and each of the other locations that you describe on pages 2-4.
Table 1. It is interesting that the soil samples contain less than half the Hg content of the sediment samples, but contain much more As. Any ideas?
Line 341. These locations are meaningless without a map.
Line 339. These lithologies are meaningless without a map.
Lines 348 through 350. Instead of speculating about the mineralogy of the samples why not take a look at the mineralogy of the samples with a petrographic microscope or analyze them with XTD? A characteristic of soils is elevated clay content. Did you examine any clay minerals?
Line 398 The best way to determine provenance is to simply look upstream at the geology; but you need a geologic map.
Author Response
Dear Sir,
Please see the attachment

Round 2
Reviewer 1 Report
line 29 - ...Upper Continental Crust (UCC) composition.
64 - ... Co and Ni are shown to be ....
113- paralava,...
130 - mainly ...
297 - elements were shown to be ...
323 - Aluminum oxide in sediment showed an average...
334 = presence of dolomite.
412 - ... can be substituted by Sr2+ ions in ....
426 - ... are shown to be closer...
467 - ... the processed solids were scattered....
508 - The REE distribution was shown to be...
512 - ... were shown to be ...
517 - The presented data should be seen...
Reviewer 2 Report
After carefully reading the revised manuscript "Geochemical Features of Major and Trace Elements ..." by Abdushukurov and others I am still disappointed. There are only a few good reasons why anyone would chemically analyze 116 randomly distributed samples of soil and underlying sediment from a large river basin. One would be to study the geochemical development of the soil over the sediments, however this was not done. Another would be to determine the provenance of the sediments. This was attempted by the authors but none of the potential source rocks were analyzed and no geologic map with potential source rocks was provided, making such an attempt impossible. Still another reason would be an attempt to determine the extent to which the soil was contaminated or degraded by "anthropic influence" presumably including any local sources of toxins such as local mining activity. This was a principal objective of the authors. However, to accomplish such an objective it would be necessary to compare the composition of samples collected near or upstream from any of the several the mines located in the valley with samples collected further away or downstream from the mines and then contour the concentration of any potential toxin around potential sources. No such comparisons were made, instead the authors conclude that with minor exceptions the composition of the soils is about the same as the sediments and that with rare exception "in very few places ..." (line 513) the composition of the soils and sediment were controlled by variations in the lithology of the bedrock. However, no attempt to identify and describe any of the "very few places" was made.
Finally, despite my clear requests there are no descriptions of the mineralogy or texture or thickness of the soils or sediments. Some soils are described as "sierozem" but none were described on the basis of the commonly used Unified Soil Classification System and no explanation of how the soil layers were distinguished from the sediment layers was made. How was the soil layer identified? It is also not clear why the top layers of soil and sediments were removed from the sample columns (line 192). If you are looking for "anthropic influence" (pollution) why remove the layer that is most likely polluted? On line 186 the authors state that they made an effort "To avoid any recent anthropic influence... " So what was the point of the whole study? I carefully looked for it but did not see it.
Round 3
Reviewer 2 Report
The authors have added a few actual sources of pollution (about two line of mine descriptions) to the conclusions which goes a long way toward supplying some meaning or purpose to the manuscript but it is still dreadfully lacking. Apparently all the interesting conclusions (the ecological assessment) of the region were already published. So we are left with a background description of the chemical variations of a collection of unconsolidated samples. No technical description distinguishing the soil samples from the sediment samples is provided, so it is unclear what the two groups of analytical data represent. The authors need to take a fresh look at the manuscript and ask the question "why would anyone (soil scientist, sedimentary petrologist, ecologist) find our manuscript interesting and important?" and then revise it accordingly.